# Determining organizational resilience through employee resilient characteristics and supportive HR practices: The moderating effect of managerial resilience

Abdulrahman Awadh Aljuaid *

Department of Human Resources Management, College of Business, University of Jeddah, Jeddah, Saudi Arabia

* aaaljuaid@uj.edu.sa

## Abstract

Resilience policies enable organizations to confront with adverse circumstances and ensure businesses operational continuity. Nevertheless, so far literature lacks how employee coping mechanism and supportive HR practices foster organizational resilience. Therefore, current study establishes research framework that combines factors such as employee resilience, psychological safety, problem focused coping, emotion focused coping, supportive human resource practices and workplace friendship and examine impact of these factors on organizational resilience. Similarly, this study has conceptualized moderating effect of managerial resilience between organizational resilience and organizational performance. The quantitative research design is taken into consideration. Data are collected through structured survey questionnaire. Overall, 421 respondents have participated in this organizational resilience research survey. These numerical responses are tested with structural equation modeling approach. Statistical results indicate that employee resilience, psychological safety, problem focused coping, emotion focused coping, supportive human resource practices and workplace friendship have depicted $R^2$= 45.4% variance in organizational resilience. Therefore, constructs like organizational resilience and managerial resilience have shown $R^2$=55.8% variance in organizational performance. Therefore, IPMA analysis has disclosed that emotion focused coping, employee resilience and managerial resilience are found most influential factors in measuring organizational resilience. This study has conceptualized employee coping mechanism towards organizational resilience and hence largely contributes to organizational resilience literature. To practice this study has suggested that factors such as employee resilience, psychological safety, problem focused coping, emotion focused coping and supportive human resource practices are crucial factors in improving organizational resilience and hence need policymakers attention. More precisely managers could enhance organizational resilience by improving employee emotion focused coping,

**Data availability statement:** All relevant data are within the manuscript and its Supporting Information files.

**Funding:** The authors extend their appreciation to the Deputyship for Research & Innovation, Ministry of Education in Saudi Arabia for funding this research work through the project number MoE-IF-UJ-R2-22-0410780-1, awarded to A.A.A. (Abdulrahman Awadh Aljuaid). The funder had no role in study design, data collection and analysis, decision to publish, or preparation of the manuscript.

**Competing interests:** The authors have declared that no competing interests exist.

employee resilience and managerial resilience. This research is original as it develops an amalgamated model that combines employee coping mechanism and supportive HR practices altogether to investigate organizational resilience. Likewise, the scope of this study is also ample as it collects observations from managers at border scale and test research framework with numerical data.

## 1 Introduction

The rising global uncertainty, natural disasters, pandemic waves, political and financial crisis have compelled organizations to introduce resilient strategies to confront adverse circumstances [1]. The term resilience denotes to the persistent of system and their ability to absorb shocks and ensure continuity of the relationship. Similarly, in organizations setting resilience is defined as organizational ability to respond quickly to external threats, absorb changes and disturbance, adapt and recover fast to maintain continuity of the business operations [1]. Organizational resilience has gained popularity among academicians and policy maker's during COVID-19 pandemic crisis. Studies have proven that organizations with resilient characteristics have remained successful to confront COVID-19 pandemic crisis [2,3,4]. Author like Stachowiak and Pawłyszyn [5] have stated that quick response to adversity and adaptability to recover are key challenges for managers. Therefore, organizations with resilience characteristics have shown more agility in business operations during turbulence [6]. Although organizational resilience is studied in the context of agility and resilience and organizational maturity however limited literature is existed that established connection between organizational resilience and employee coping mechanism [7,3].

Prior studies have discussed ways to improve organizational resilience through employee well-being and high performance work system Abboh et al. [8]; Ali et al. [9]; Rahi [10] nevertheless relationship between organizational resilience and supportive human resource practices is yet to be examined. Likewise, organizational resilience is rarely measured with factors such as managerial resilience, employee resilience, psychological safety and workplace friendship. To fill this research gap current study has schematized that factors such as employee resilience, psychological safety, problem focused coping, emotion focused coping, supportive HR practices and workplace friendship are core factors which foster organizational resilience in adverse situation. Following that, current study has examined organizational resilience issue with four core research objectives. To understand how employee resilience and employee psychological safety influence organizational resilience? To examine how problem focused coping and emotion focused coping impact organizational resilience? To examine how supportive HR practices and workplace friendship influence organizational resilience? To examine how managerial resilience moderates the relationship between organizational resilience and organizational performance? Thus, the scope of this study is large as it collects observations from managers at border scale and test research framework with

numerical data. The findings of this research assist managers in recognizing factors which enhance organizational resilience and organizational performance in adverse situation. The remaining of this research is followed by literature review, research methodology, data analysis, discussion, research contributions, conclusion, research limitations and future research directions.

## 2 Literature review

### 2.1 Employee resilience and psychological safety

Organizations are continuously striving to find out solutions to deal with uncertainty and unprecedented events. Therefore, recent studies have recognized resilience an important factor that reduces employee stress during uncertainty and motivate employee to deal with turbulence [11,12]. The term organizational resilience is referred to firm ability to deal with vulnerability, reformulating business strategies and ensure operational continuity during disruption. Although organizational resilience is evaluated with some renowned factors such as high performance work system, work engagement and psychological resilience Hanu and Khumalo [13]; Ibrahim and Hussein [14]; P. Kim et al. [6] however less attention is paid on employee resilience. The term employee resilience is the extent wherein employees are capable to respond actively to disruption, adapt and develop new polices to deal with uncertainty. Prior studies have shown clear evidence that employee resilience is positively related to organizational resilience and therefore organizations should promote resilient attributes among employees to achieve organizational resilience and performance [11,15–18]. Another important factor is employee psychological safety that matters a lot during disruption. The term psychological safety is the extent wherein employee gets opportunity to share ideas, take risks and freely express their thoughts during unpredictable situation [18–20]. Studies have also shown that employee psychological safety is positively related to organizational resilience [21,12]. Therefore, following hypotheses are put forward:

H1: Employee resilience is positively related to organizational resilience.

H2: Psychological safety is positively related to organizational resilience.

### 2.2 Problem and emotion focused coping

The coping mechanism is recognized key antecedent of individual behavior to manage stress and external threats [11]. Therefore, organizations have dealt with crisis through coping strategies [22,7,23]. The notion of is seen as individual cognitive and behavioral efforts to regulate employee emotions and thoughts in response to stressful events [24,13]. In literature coping concept is further discussed either as problem focused or emotion focused [14,25]. The problem focused coping deals with problems, change organizational environment through interpersonal communication and bring problem solving strategies at workplace. Therefore, emotion focused coping denotes to manage emotional distress in the face of uncertainty and disruption [26,23]. Emotional focused coping involves broader range of emotions including rejection, alienation, venting of emotions, avoidance and positive interpretation. Nevertheless, in this study it is assumed that both problem focused coping and emotion focused coping will bring positive change in organizational operations and hence boost organizational resilience. Employees with problem focused coping are found in better position to respond environmental changes [18,27,26]. Similarly, studies have also established that emotion coping enables employees to get control on emotions and eliminate stress during uncertainty and encourage confronting unprecedented situation [6,21,28]. Thus, following hypotheses are assumed:

H3: Problem focused coping is positively related to organizational resilience.

H4: Emotion focused coping is positively related to organizational resilience.

## 2.3 Supportive HR practices and workplace friendship

There are some other factors that build organizational resilience for instance supportive human resource practices and workplace friendship. Both factors have substantial support from prior literature in measuring organizational resilience [2,29,30]. Although the role of traditional human resource practices is vigilant in achieving organizational performance Jo et al. [27]; Naqshbandi et al. [22] however limited literature is existed in the context of supportive human resource practices. Supportive human resource practices are quite different from traditional human resource practices and focused on need based scenarios. Supportive human resource practices are defined as practices that give recognition to employees, embodied emotional support, care, respect, feeling of honor and instrumental support to confront with unprecedented situation including disastrous events [2]. According to Zhang et al. [4] stated that under the influence of supportive human resource practices employees will pay more attention to achieve organizational goals. Therefore, present study has assumed that supportive human resource practices builds resilience attributes among employees and hence boosts organizational resilience. Moving further studies have recognized workplace friendship as unique factor that enhances organizational resilience [19,30,31]. Studies have shown that workplace friendship positively contributes to work environment resulting better work engagement and increase in organizational resilience [19,30]. Therefore, current study has projected that both supportive practices and workplace friendship boost organizational resilience. Thus, following hypotheses are put forward:

H5: Supported HR practices are positively related to organizational resilience.

H6: Workplace friendship is positively related to organizational resilience.

## 2.4 Managerial resilience

Although organizational resilience has key importance in measuring organizational performance however the role of managerial resilience cannot be ignored. Resilience denotes to the ability to adapt and recover from adversities and disruption [26]. Nevertheless, managerial resilience is explained as manager's ability to overcome high impact challenges, quick respond to unfavorable situations and ensure business continuity during disruption [11]. According to Liang and Cao [7] has stated that managers having ability to work in stressful environment will boost organizational resilience. In addition to that literature has also revealed that managerial resilience promote resilient culture through policies and encourage employees to exert their resilience during disruption [1,11,32]. Although relationship between managerial resilience and organizational performance has established in past studies Liang and Cao [7]; Liang and Li [7] nonetheless moderating effect of managerial resilience is yet to be examined. Therefore, in this study managerial resilience is conceptualized as moderating factor between organizational resilience and organizational performance and exhibited in Fig 1 Author like Liang and Cao [7] has confirmed moderation effect of managerial resilience between employee resilience and emotion focused coping. Following that notion current study has assumed that higher level of managerial resilience would strengthen the relationship between organizational resilience and organizational performance. Therefore, following hypotheses are assumed:

H7: Organizational resilience is positively related to organizational performance.

H8: Managerial resilience moderates the relationship between organizational resilience and organizational performance.

## 3 Methodology

### 3.1 Research methods and scale

The current research is grounded in positivist research paradigm and strives to identify factors which enhance organizational resilience and organizational performance. After reviewing literature this study has schematized factors like

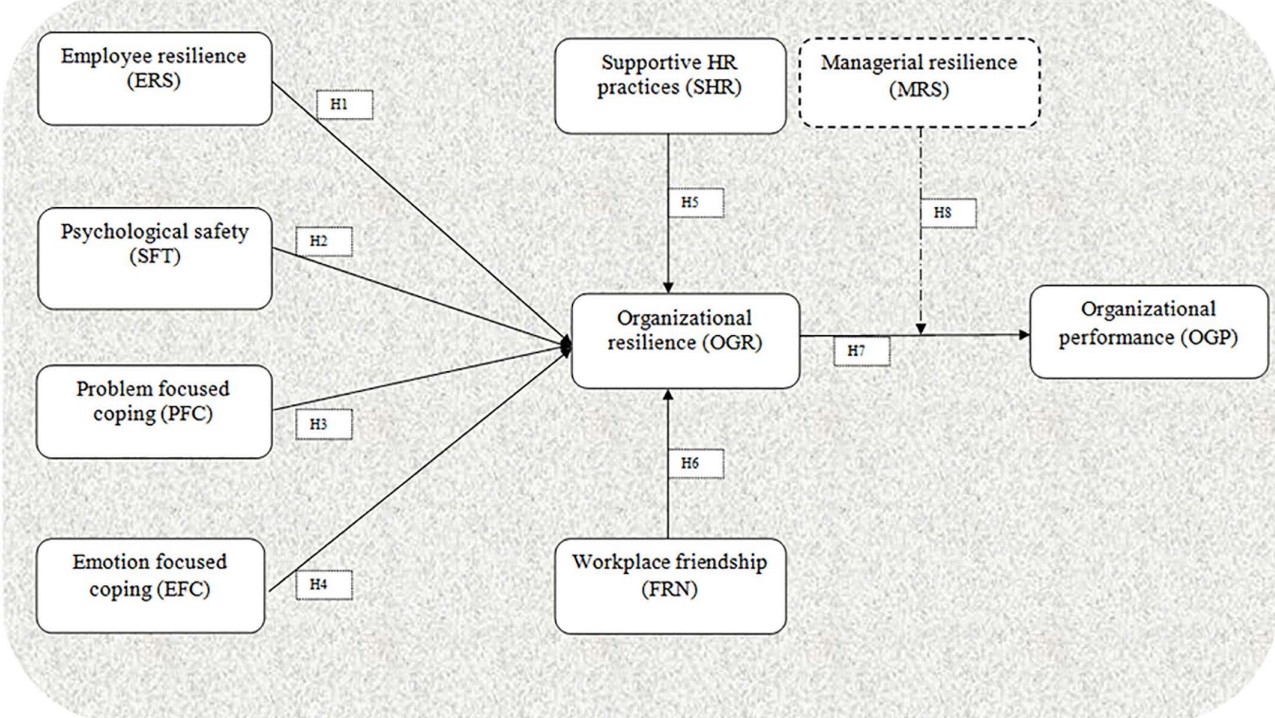

**Fig 1. Research framework.**

employee resilience, psychological safety, problem focused coping, emotion focused coping, supportive human resource practices and workplace friendship and makes connection with organizational resilience. In addition to that this study has added moderating path in research framework and conceptualized moderating effect of managerial resilience between organizational resilience and organizational performance. Therefore, research framework of this study is empirically tested followed by quantitative research approach. Nevertheless, for quantitative analysis data were collected from managers working in manufacturing organizations of Saudi Arabia. Selection of the managers is consistent with prior study conducted by Yamin et al. [33] has stated that managers have better knowledge about resilience policies and therefore considered right population in this study. Empirical data have been collected through purposive sampling approach. Selecting purposive sampling method is adequate as researcher has intentionally selected managers to participate in organizational resilience research survey and consistent with past studies [33,34]. The unit of the analysis is organizational resilience and tested with quantitative data collected through structured questionnaire.

Survey questionnaire is designed for data collection and comprised into two sections. The first section of the questionnaire comprises respondent's profile. Nevertheless, the second part of the survey questionnaire is grounded in construct items. Each construct is measured with multiple scale items adapted from past studies. Construct items for employee resilience are adapted from [11,18,35]. Therefore, instrument items for psychological safety are adapted from [19]. Concerning with coping mechanism construct items for problem focused coping are adapted from [36]. Therefore, emotion focused coping construct items are adapted from [25,6,37]. Similarly, construct items for supportive human resource practices are adapted from Zhang et al. [4] and [29]. Instrument items for the construct workplace friendship are adapted from [19]. Moving further organizational resilience items are adapted from [38]. Construct items for organizational performance are adapted from Li et al. [12] and [39]. Therefore, managerial resilience items are adapted from Liang and Cao

[7] and [40]. Within structured questionnaire scale items are labeled with Likert type scale with range of 1–7 (1 to strongly disagree and 7 to strongly agree).

### 3.2 Sampling and data collection

Though population of this study is vibrant and consistent with past study Yamin [26] nevertheless, selecting accurate sample size is critical. Therefore, researcher has followed guidelines provided by Hair et al. [41] and [10]. As per guideline provided by Rahi [39] has recommended that data must be 10 times greater than constructs indicators. Overall, this study has 28 indicators with 9 constructs and therefore required sample size is 280. Aside of that studies have revealed that sample size of 200 is considered adequate for structural equation modeling approach [42–45]. Data were collected physically during the months of August 2024. Respondents were briefed about research survey through cover letter and requested to participate in organizational resilience survey voluntarily. Overall, 455 respondents were approached however 421 respondents had returned questionnaire with valid responses. Among these 421 respondents 87.2% were male and 12.8% were female. The majority of male respondents were expected as Saudi Arabia is considered male dominating society. Concerning with respondent's professional experience 13.5% respondents have shown 1–5 years work experience. Therefore, 52% respondents have shown 6–10 years industry experience. Similarly, 34.4% respondents have shown 11–15 years of experience. Respondent's age statistics have revealed that 34% respondents were aged between 21–30 years. Next to this 40.4% respondents have age range between 31–40 years. Nevertheless, 25.7% respondents have revealed their age between 41–50 years. Finally, these numerical are tested with structural equation modeling approach. This study involved human participants who voluntarily completed an online questionnaire. Participation was entirely voluntary, and respondents were informed about the study's purpose, the confidentiality of their responses, and their right to withdraw at any time without consequences. Since the survey was conducted online, informed consent was obtained electronically before participants could proceed with the questionnaire. The questionnaire and methodology for this study were reviewed and approved by the Human Research Ethics Committee of the University of Jeddah (Ethics approval number: UJ-REC-394).

## 4 Data analysis

### 4.1 Common method bias

The numerical data are collected through structured questionnaire and therefore chances are there that data could be affected with common method bias issue. Nevertheless, this issue is addressed though known remedies like procedural and statistical remedies. At first stage CMB issue is addressed through procedural remedies. This method has suggested that questionnaire items should be jumbled up prior to distribution. Following that items were jumbled up to mitigate the risk of CMB. Concerning with statistical remedies CMB issue is addressed with Harman's single factor analysis. To ensure that CMB is not an issue the recommend threshold value is 40%. Nevertheless, data are estimated with Harman's single factor analysis. Results have disclosed that maximum variance explained by first factor is less than threshold value, i.e., 22% and hence ensured that data are valid and free from any kind of CMB issue.

### 4.2 Structural equation modeling

The structural equation modeling approach is a comprehensive statistical approach and analyzes complex data in two stages namely measurement model and structural model [46,3]. The measurement model stage confirms constructs reliability, discriminant validity and convergent validity. Therefore, in second stage hypotheses are confirmed through structural assessment. Following measurement model construct reliability is achieved with composite reliability with threshold value of.70 [40]. Therefore, convergent validity is achieved with average variance extracted with threshold value of.50 [40]. Nevertheless, indicator reliability is achieved following threshold values of.60 [40]. Data are analyzed and revealed

satisfactory indicator reliability, convergent validity and constructs reliability. Refined results of the measurement model are exhibited in Table 1.

Although constructs reliability is achieved however discriminant validity is yet to be assessed. Discriminant validity ensures that constructs measure unique concept and could be measured with cross loading methods, HTMT or Fornell and Larcker analysis. All these methods have substantial support from prior literature and widely recommended [42,46,47]. Following guidelines provided by Fornell [48] data are analyzed. To ensure discriminant validity average variance extracted values are measured. This analysis has suggested threshold value of average variance extracted on square root must be greater. Nevertheless, statistical results are shown in Table 2 have revealed adequate values of average variance extracted and hence confirmed discriminant validity of the constructs.

Data are analyzed with Heterotrait-monotrait ratio of correlations to ensure discriminant validity of the construct. The HTMT ratio analysis is an advance test and ensure discriminant validity of the constructs [47]. This method has recommended that HTMT ratio must be lower than.85 to confirm discriminant validity of the constructs [47,49]. Results of the HTMT analysis have revealed that none of the HTMT value was greater than threshold value, i.e.,.85 and hence these findings have confirmed discriminant validity of the factors. Table 3 depicts results of the HTMT analysis with satisfactory HTMT values.

**Table 1. Measurement model.**

| Scale items | Loadings | alpha | CR | AVE |
|---|---|---|---|---|
| EFC1: Employee tries to forget disruption and confront with crisis. | 0.776 | 0.816 | 0.879 | 0.645 |
| EFC2: Employee refuses to think about destruction and continue work. | 0.833 | | | |
| EFC3: Employee can control on emotions during disaster. | 0.791 | | | |
| EFC4: Intensity of the catastrophic events cannot distract employee. | 0.811 | | | |
| ERS1: I always take stressful things at workplace. | 0.890 | 0.836 | 0.901 | 0.753 |
| ERS2: I have ability to adapt during disruption. | 0.829 | | | |
| ERS3: I accept challenges at workplace in adverse circumstances. | 0.883 | | | |
| FRN1: Employees in this organization are sociable with coworkers | 0.916 | 0.755 | 0.890 | 0.802 |
| FRN2: Coworkers strong relationship motivates me to continue job. | 0.874 | | | |
| MRS1: Managers are capable to deal with catastrophic events. | 0.790 | 0.724 | 0.843 | 0.642 |
| MRS2: Managers have ability to forecast disruption in advance. | 0.796 | | | |
| MRS3: Managers have creative solutions to alter difficult situation. | 0.818 | | | |
| OGP1: Our organization is offering promising services. | 0.881 | 0.851 | 0.910 | 0.770 |
| OGP2: Our organization has broad market prospect. | 0.890 | | | |
| OGP3: Our organization has strong competitive advantage. | 0.861 | | | |
| OGR1: This organization has adaptive capacity to deal with crisis. | 0.850 | 0.844 | 0.905 | 0.761 |
| OGR2: Organization is able to confront with unexpected situation. | 0.881 | | | |
| OGR3: Organization has capacity to absorb external shocks. | 0.885 | | | |
| PFC1: Disturbance in operations motivates me to pay more attention. | 0.866 | 0.672 | 0.859 | 0.753 |
| PFC2: Disruptive events bring new opportunities to solve problem. | 0.869 | | | |
| SFT1: I believe that people around me are not trouble makers. | 0.825 | 0.757 | 0.861 | 0.673 |
| SFT2: During crisis I get complete psychological support from peers. | 0.843 | | | |
| SFT3: Employees get complete support to resolve problem. | 0.793 | | | |
| SHR1: Employees get reward through performance appraisal. | 0.831 | 0.833 | 0.888 | 0.666 |
| SHR2: Employees get equal growth opportunities in this organization. | 0.814 | | | |
| SHR3: I get equally opportunity in decision making process. | 0.794 | | | |
| SHR4: During disruption employees are allowed to take key decisions. | 0.824 | | | |

**Table 2. Fornell and Larcker analysis.**

| Factors | EFC | ERS | FRN | MRS | OGP | OGR | PFC | SFT | SHR |
|---|---|---|---|---|---|---|---|---|---|
| EFC | 0.803 | | | | | | | | |
| ERS | 0.355 | 0.868 | | | | | | | |
| FRN | 0.247 | 0.552 | 0.895 | | | | | | |
| MRS | 0.285 | 0.285 | 0.168 | 0.802 | | | | | |
| OGP | 0.539 | 0.527 | 0.367 | 0.235 | 0.878 | | | | |
| OGR | 0.612 | 0.463 | 0.369 | 0.265 | 0.727 | 0.872 | | | |
| PFC | 0.354 | 0.204 | 0.249 | 0.338 | 0.328 | 0.373 | 0.868 | | |
| SFT | 0.408 | 0.281 | 0.246 | 0.371 | 0.398 | 0.398 | 0.441 | 0.820 | |
| SHR | 0.334 | 0.691 | 0.543 | 0.289 | 0.485 | 0.473 | 0.230 | 0.302 | 0.816 |

**Table 3. HTMT analysis.**

| | EFC | ERS | FRN | MRS | OGP | OGR | PFC | SFT | SHR |
|---|---|---|---|---|---|---|---|---|---|
| EFC | | | | | | | | | |
| ERS | 0.427 | | | | | | | | |
| FRN | 0.313 | 0.696 | | | | | | | |
| MRS | 0.368 | 0.368 | 0.223 | | | | | | |
| OGP | 0.640 | 0.620 | 0.454 | 0.296 | | | | | |
| OGR | 0.729 | 0.544 | 0.460 | 0.333 | 0.842 | | | | |
| PFC | 0.478 | 0.272 | 0.354 | 0.486 | 0.429 | 0.491 | | | |
| SFT | 0.516 | 0.353 | 0.326 | 0.499 | 0.494 | 0.495 | 0.617 | | |
| SHR | 0.398 | 0.829 | 0.688 | 0.370 | 0.575 | 0.560 | 0.307 | 0.378 | |

Aside of HTMT cross loading analysis is conducted to ensure the discriminant validity of the constructs. This method evaluates discriminant validity using indicator loadings. Data were analyzed to disclose results. Findings of the cross loading analysis has indicated that loadings are satisfactory and less than corresponding construct loadings. Therefore, adequate loadings of the indicators have confirmed that constructs are discriminant and valid for structural assessment. Cross loadings values are shown in Table 4 wherein each construct loading is higher when comparing with corresponding construct loadings.

### 4.3 Hypotheses analysis

The second stage of the structural equation modeling incorporates process of the bootstrapping to disclose t-statistics and path significance. Bootstrapping in hypotheses analysis is highly recommended as it mitigates data normality issue. Data were bootstrapped and results demonstrate positive impact of employee resilience in determining organizational resilience and confirmed H1 supported by $\beta = 0.258$, SE 0.066, t-statistics 3.904 $p < 0.000$. Psychological safety is positively related to organizational resilience supported by $\beta = 0.114$, SE 0.058, t-statistics 1.985 $p < 0.025$ and hence confirmed H2. Similarly, problem focused coping has shown positive influence in organizational resilience and established H3 supported by $\beta = 0.073$, SE 0.040, t-statistics 1.817 $p < 0.036$. Likewise, emotion focused coping is positively related to organizational resilience supported by $\beta = 0.324$, SE 0.048, t-statistics 6.781 $p < 0.000$ and hence confirmed H4. Moreover, there is positive relationship between supportive human resource practices and organizational resilience and hence confirmed H5 supported by $\beta = 0.134$, SE 0.062, t-statistics 2.169 $p < 0.016$. Nevertheless, contrary to researcher expectations workplace friendship has shown negative impact organizational resilience and hence H6 is rejected due to insignificant values of $\beta = 0.026$, SE 0.055, t-statistics 0.463 $p < 0.322$. However,

**Table 4. Cross loadings analysis.**

| Factors | EFC | ERS | FRN | MRS | OGP | OGR | PFC | SFT | SHR |
|---|---|---|---|---|---|---|---|---|---|
| EFC1 | 0.776 | 0.274 | 0.201 | 0.293 | 0.390 | 0.438 | 0.277 | 0.313 | 0.258 |
| EFC2 | 0.833 | 0.289 | 0.222 | 0.235 | 0.424 | 0.451 | 0.325 | 0.347 | 0.250 |
| EFC3 | 0.791 | 0.253 | 0.147 | 0.163 | 0.426 | 0.485 | 0.236 | 0.303 | 0.225 |
| EFC4 | 0.811 | 0.320 | 0.222 | 0.231 | 0.481 | 0.577 | 0.299 | 0.346 | 0.330 |
| ERS1 | 0.325 | 0.890 | 0.501 | 0.263 | 0.434 | 0.402 | 0.175 | 0.240 | 0.608 |
| ERS2 | 0.277 | 0.829 | 0.495 | 0.244 | 0.431 | 0.429 | 0.178 | 0.243 | 0.611 |
| ERS3 | 0.322 | 0.883 | 0.447 | 0.239 | 0.500 | 0.380 | 0.178 | 0.247 | 0.583 |
| FRN1 | 0.227 | 0.513 | 0.916 | 0.156 | 0.358 | 0.337 | 0.200 | 0.215 | 0.483 |
| FRN2 | 0.215 | 0.474 | 0.874 | 0.143 | 0.295 | 0.324 | 0.252 | 0.227 | 0.492 |
| MRS1 | 0.177 | 0.224 | 0.105 | 0.790 | 0.187 | 0.178 | 0.280 | 0.282 | 0.213 |
| MRS2 | 0.259 | 0.258 | 0.154 | 0.796 | 0.184 | 0.216 | 0.285 | 0.302 | 0.251 |
| MRS3 | 0.241 | 0.207 | 0.138 | 0.818 | 0.194 | 0.235 | 0.252 | 0.306 | 0.229 |
| OGP1 | 0.468 | 0.477 | 0.326 | 0.194 | 0.881 | 0.638 | 0.257 | 0.335 | 0.444 |
| OGP2 | 0.522 | 0.479 | 0.328 | 0.252 | 0.890 | 0.688 | 0.357 | 0.362 | 0.429 |
| OGP3 | 0.422 | 0.429 | 0.311 | 0.166 | 0.861 | 0.582 | 0.241 | 0.351 | 0.404 |
| OGR1 | 0.530 | 0.330 | 0.283 | 0.209 | 0.535 | 0.850 | 0.309 | 0.331 | 0.401 |
| OGR2 | 0.491 | 0.398 | 0.365 | 0.247 | 0.616 | 0.881 | 0.298 | 0.355 | 0.415 |
| OGR3 | 0.576 | 0.467 | 0.315 | 0.235 | 0.727 | 0.885 | 0.363 | 0.354 | 0.420 |
| PFC1 | 0.295 | 0.182 | 0.199 | 0.278 | 0.283 | 0.286 | 0.866 | 0.365 | 0.216 |
| PFC2 | 0.320 | 0.172 | 0.233 | 0.309 | 0.286 | 0.361 | 0.869 | 0.401 | 0.183 |
| SFT1 | 0.364 | 0.220 | 0.204 | 0.298 | 0.353 | 0.348 | 0.382 | 0.825 | 0.259 |
| SFT2 | 0.302 | 0.217 | 0.231 | 0.263 | 0.311 | 0.302 | 0.315 | 0.843 | 0.262 |
| SFT3 | 0.335 | 0.254 | 0.171 | 0.353 | 0.312 | 0.326 | 0.387 | 0.793 | 0.222 |
| SHR1 | 0.234 | 0.604 | 0.502 | 0.230 | 0.379 | 0.378 | 0.188 | 0.221 | 0.831 |
| SHR2 | 0.342 | 0.577 | 0.395 | 0.257 | 0.427 | 0.451 | 0.221 | 0.294 | 0.814 |
| SHR3 | 0.270 | 0.497 | 0.414 | 0.242 | 0.381 | 0.353 | 0.226 | 0.245 | 0.794 |
| SHR4 | 0.235 | 0.574 | 0.466 | 0.213 | 0.393 | 0.354 | 0.115 | 0.222 | 0.824 |

organizational resilience has depicted positive impact organizational resilience and supported by β = 0.700, SE 0.032, t-statistics 22.217 p < 0.000 and hence H7 is established. Table 5 depicts results of the hypotheses with path significance and t-statistics.

### 4.4 Importance performance matrix

The research framework of this study has multiple factors and therefore clarity is required for policymakers to understand importance of outlined factors. In this essence importance performance matrix analysis is performed by selecting organizational performance as an outcome factor. Data are analyzed with importance performance matrix algorithm. Results indicate that organizational resilience is the most impactful factor to predict organizational performance. Next to this emotion focused coping is considered the second most important factor to determine organizational performance. Similarly, employee resilience and managerial resilience have revealed medium level of importance to determine organizational performance. However, workplace friendship has revealed least importance. Overall, results have revealed that organizational resilience, emotion focused coping, employee resilience and managerial resilience are found influential factors in measuring organizational performance. Table 6 depicts results of the IPMA analysis with each factor importance and performance.

**Table 5. Hypothesis analysis.**

| Hypothesis | Path | β | STDEV | t-statistics | Significance | Decision |
|---|---|---|---|---|---|---|
| H1 | ERS -> OGR | 0.258 | 0.066 | 3.904 | 0.000 | Accepted |
| H2 | SFT -> OGR | 0.114 | 0.058 | 1.985 | 0.025 | Accepted |
| H3 | PFC -> OGR | 0.073 | 0.040 | 1.817 | 0.036 | Accepted |
| H4 | EFC -> OGR | 0.324 | 0.048 | 6.781 | 0.000 | Accepted |
| H5 | SHR -> OGR | 0.134 | 0.062 | 2.169 | 0.016 | Accepted |
| H6 | FRN -> OGR | 0.026 | 0.055 | 0.463 | 0.322 | Rejected |
| H7 | OGR -> OGP | 0.700 | 0.032 | 22.217 | 0.000 | Accepted |

**Table 6. Importance performance analysis.**

| Factors | Importance impact | Performance impact |
|---|---|---|
| Emotion focused coping | 0.227 | 68.036 |
| Employee resilience | 0.181 | 71.246 |
| Workplace friendship | 0.018 | 68.291 |
| Managerial resilience | 0.111 | 72.793 |
| Organizational resilience | 0.700 | 68.013 |
| Problem focused coping | 0.051 | 68.179 |
| Psychological safety | 0.080 | 70.890 |
| Supportive HR practices | 0.094 | 73.059 |

## 4.5 Coefficient of determination $R^2$ and effect size analysis $f^2$

The collective impact of the underpinned constructs is assessed with coefficient of determination, i.e., $R^2$. Results as shown in S1 Appendix demonstrated that collectively employee resilience, psychological safety, problem focused coping, emotion focused coping, supportive human resource practices and workplace friendship have depicted $R^2=45.4\%$ variance in organizational resilience. Therefore, constructs like organizational resilience and managerial resilience have shown $R^2=55.8\%$ variance in organizational performance. Similarly, constructs effect size is evaluated with $f^2$ following threshold value of.02, 15 and.35 representing small, medium and large effect size respectively. Results of the effect size analysis have disclosed small effect size of all exogenous construct in measuring outcome construct. However, organizational resilience has shown large effect size in determining organizational performance. Results of the effect size analysis are shown in Table 7 with both outcome constructs organizational resilience and organizational performance.

## 4.6 Moderating analysis

The research model has theorized moderating relationship of managerial resilience between organizational resilience and organizational performance. Therefore, statistical analysis is conducted with product indicator approach. Data are bootstrapped to disclose significance of the moderating path. Results of the moderating analysis have explained significant moderating impact of managerial resilience between organizational resilience and organizational performance supported by β = 0.137, SE 0.047, t-statistics 2.889 significant at 0.002 and thus H8 is established. In addition to that trend of the moderation is tested with simple slope analysis. Simple slope gradient has depicted sharp and upward trend at MRS at + ISD. On the flip side MAR at – ISD is depicting downward trend. These findings have established that with the presence of managerial resilience relationship between organizational resilience and organizational performance will be stronger. Fig 2 depicts moderation relationship strength at positive and negative gradient.

Table 7. Effect size analysis ($f^2$).

| Factors | Results | Effect size |
|---|---|---|
| *Organizational resilience* | | |
| Emotion focused coping | 0.141 | Small |
| Employee resilience | 0.057 | Small |
| Workplace friendship | 0.001 | Small |
| Problem focused coping | 0.007 | Small |
| Psychological safety | 0.017 | Small |
| Supportive HR practices | 0.016 | Small |
| *Organizational performance* | | |
| Managerial resilience | 0.026 | Small |
| Organizational resilience | 1.045 | Substantial |

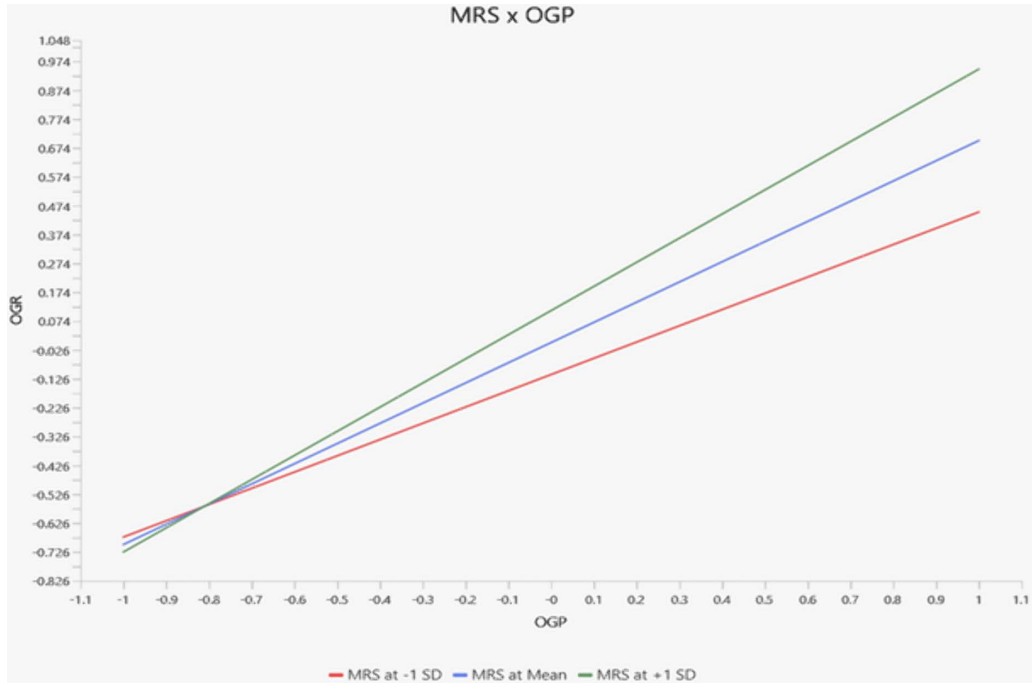

Fig 2. Simple slope gradient.

## 5 Discussion

Organizational resilience has gained policymakers attentions in last few years due to rise in environmental changes and disastrous events. According to Kunz and Sonnenholzner [1] have postulated that resilience strategies enabled organizations to face adverse circumstance proactively and must be implemented in organizations. Though past studies have examined organizational resilience Liang and Cao [7]; Stachowiak and Pawłyszyn [3] however employee and managerial perspective is yet to be investigated. Moreover, there is strong evidence in literature that executive traits exert positive impact on organizational abilities and capable organizations to take prompt decision during turbulence [1]. Consistent with above argument and grounded in Upper Echelon Theory this study has developed organizational resilience research framework. The research model comprises factors such as employee resilience, psychological safety, problem focused

coping, emotion focused coping, supportive human resource practices, workplace friendship and managerial resilience to investigate organizational resilience and organizational performance. With the help of literature hypotheses are established and then tested with structural equation modeling approach. Results are compared and contrast with past studies. For instance employee resilience has revealed positive impact in measuring organizational resilience and consistent with past studies [7,12]. Psychological safety is positively related to organizational resilience and in line with past studies [21,12,45].

Results of this study are further compared with past studies and revealed that problem focused coping has shown positive influence in organizational resilience and endorsing arguments established by past studies [7,50,23]. Employee emotion focused coping is positively related to organizational resilience and in line with [18,27,26]. Moving further this study has confirmed supportive human resource practices are positively impact organizational resilience and consistent with past studies [9,50,4]. Interestingly, workplace friendship has shown negative impact in measuring organizational resilience and hence this finding is inconsistent with past studies [21,50]. This could be happened because close friendships to create groupthink, reduce constructive conflict, or lead to blurred professional boundaries that may hinder resilience during crises. Therefore, one can infer that workplace friendship is not necessary to improve organizational resilience. Results are further examined and found significant moderating impact of managerial resilience between organizational resilience and organizational performance and hence confirming arguments established by past studies [2,7,37]. Overall, this study has revealed substantial variance $R^2$=45.4% in measuring organizational resilience and $R^2$=55.8% in organizational performance and hence this impact is considerable and attracts policymakers. Although impact of all exogenous constructs except workplace friendship is found positive in measuring organizational resilience however importance of the constructs is reviewed with IPMA analysis. Therefore, current study has summarized that in terms of importance factors like emotion focused coping, employee resilience and managerial resilience are found most influential factors in measuring organizational resilience and hence these factors must be part of the organizational strategic planning.

## 6 Research contributions

### 6.1 Theoretical contributions

This study has numerous contributions to theory and largely enriches resilience literature. For instance this study has disclosed two core dimensions of employee coping mechanism namely problem focused coping and emotion focused coping and hence contributes to literature. Moreover, conceptualizing relationship of employee coping mechanism towards organizational resilience is also valuable and hence contributes to organization resilience literature. Although prior studies have explored organizational resilience with factors such as corporate sustainability and resilient leadership Garrido-Moreno et al. [39]; Prayag et al. [12] however, literature is scarce on organizational resilience with factors such as employee resilience, psychological safety and workplace friendship. Therefore, predicting organizational resilience with factors such as employee resilience, psychological safety and workplace friendship is unique and hence contributes to literature. Another unique contribution of this study is to include moderating factor into research framework. This study has added managerial resilience as moderating factor between resilience and organizational performance and hence substantially contributed to resilience literature. Overall, this study has revealed that employee resilience, psychological safety, problem focused coping, emotion focused coping, supportive human resource practices and workplace friendship have explained $R^2$=45.4% variance in organizational resilience and hence ensure the theoretical validity of the research model. Similarly, constructs like organizational resilience and managerial resilience have shown $R^2$=55.8% variance in organizational performance and hence confirmed theoretical validity of the research model with substantial coefficient of determination. Moreover, amalgamation of employee resilience, psychological safety, problem focused coping, emotion focused coping, supportive human resource practices, workplace friendship and managerial resilience into one research framework and examining organizational resilience has largely contributed to literature in the context of organizational performance.

## 6.2 Practical contributions

Like theoretical contributions this study has numerous useful practical contributions. In macro perspective this study has established that resilient framework could help organizations to sustain in turbulent environment and boost organizational performance. More precisely this study has revealed that factor such as psychological safety and employee resilience are influential factors and enhance organizational resilience. This finding has directed that policymakers should pay attention in improving employee psychological safety and employee resilience which in turn increase employee ability to face catastrophic events. Similarly, this study has revealed positive influence of problem focused coping and emotion focused coping in measuring organizational resilience. This indicate that improving employee problem focused coping and emotion focused coping will boost employee confidence to continue work during disruption and hence improve organizational resilience and performance. Moreover, this study has suggested that supportive human resource practices bring flexibility at workplace, empower employees to take prompt decision and hence increase organizational resilience. Therefore, policymakers should familiarize supportive human resource practices at workplace which in fact encourage employee to react promptly to turbulence and increase organizational resilience. Aside of employee resilience the role of managerial resilience is found critical in improving organizational resilience and performance. Nevertheless, moderating effect of managerial resilience has revealed that higher managerial resilience will stronger the relationship between organizational resilience and organizational performance. Therefore, policymakers should train managers to confront external changes resulting better organizational resilience and organizational performance during crisis. Likewise, IPMA analysis has disclosed that emotion focused coping, employee resilience and managerial resilience are found most influential factors in measuring organizational resilience and hence need policy maker's attention. Thus, continuity of business operations in the face of disruption is possible with strong employee emotion focused coping, employee resilience and managerial resilience and therefore policymakers must consider these factors while developing new policies in organizations.

## 6.3 Conclusion

This study aims to uncover black box between employee coping mechanism and organizational resilience. Therefore, research framework is developed that combines factors such as employee resilience, psychological safety, problem focused coping, emotion focused coping, supportive human resource practices and workplace friendship to investigate organizational resilience. Results indicate that employee resilience, psychological safety, problem focused coping, emotion focused coping, supportive human resource practices and workplace friendship have shown $R^2$=45.4% variance in organizational resilience. Similarly, constructs like organizational resilience and managerial resilience have shown $R^2$=55.8% variance in organizational performance. Moreover, statistical findings have established that with the presence of managerial resilience relationship between organizational resilience and organizational performance will be stronger. In term of contributions this study has disclosed two core dimensions of employee coping mechanism namely problem focused coping and emotion focused coping and hence largely contributes to literature. Similarly, this study has added managerial resilience as moderating factor between resilience and organizational performance and hence substantially contributes to resilience literature. For policymakers this study has suggested that factors such as employee resilience, psychological safety, problem focused coping, emotion focused coping and supportive human resource practices are crucial factors in improving organizational resilience and hence need policymakers attention. Nevertheless, IPMA analysis has identified that emotion focused coping, employee resilience and managerial resilience are most influential factors. These findings indicate that managers could foster organizational resilience by improving employee emotion focused coping, employee resilience and managerial resilience. This study is unique as it develops an amalgamated model that combines employee coping mechanism and supportive HR practices altogether and investigates organizational resilience.

### 6.4 Limitations and future directions

This study has some limitations and therefore acknowledged for future research directions. Although research framework has included core factors which enhance organizational resilience however this study does not guarantee to add all factors which impact organizational resilience. Factors such as employee psychological empowerment and employee voice could enhance organizational resilience. Therefore, future researchers are suggested to extend current research model some additional factors. Similarly, this study has established research model with resilience theory and upper echelon theory however future researchers are suggested to extend current research model with some other renowned theories like theory of planned behavior TPB and technology organization environment TOE framework. Another limitation of this study is that this study is conducted in developing region and therefore outcome may vary in developed region. Therefore, future researchers are suggested to conduct comparative research and reveal differences that how organizational resilience varies between developing and developed region. As this study is cross sectional and hence included respondents opinion at one point in time. Nevertheless, future researchers are suggested to conduct longitudinal research to get in-depth insight of organizational resilience. Finally, future researchers are suggested to collect data from front desk employees and reveal how managers and employees opinion differ towards organizational resilience.

## Supporting information

**S1 File. IPMA HRM.**
(XLSX)

**S2 File. PLS Algorithm HRM.**
(XLSX)

**S3 File. Bootstrapping HRM.**
(XLSX)

**S4 File. PLS Predict HRM.**
(XLSX)

**S1 Appendix. Path and Coefficient of determination $>R^2$.**
(DOCX)

**S1 Data. Data.**
(XLSX)

## Author contributions

**Conceptualization:** Abdulrahman Awadh Aljuaid.

**Data curation:** Abdulrahman Awadh Aljuaid.

**Formal analysis:** Abdulrahman Awadh Aljuaid.

**Funding acquisition:** Abdulrahman Awadh Aljuaid.

**Investigation:** Abdulrahman Awadh Aljuaid.

**Methodology:** Abdulrahman Awadh Aljuaid.

**Project administration:** Abdulrahman Awadh Aljuaid.

**Resources:** Abdulrahman Awadh Aljuaid.

**Software:** Abdulrahman Awadh Aljuaid.

**Supervision:** Abdulrahman Awadh Aljuaid.

**Validation:** Abdulrahman Awadh Aljuaid.

**Visualization:** Abdulrahman Awadh Aljuaid.

**Writing – original draft:** Abdulrahman Awadh Aljuaid.

**Writing – review & editing:** Abdulrahman Awadh Aljuaid.

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
