## [Decision Letter · Decision Letter 0]

24 Jun 2025

Dear Dr. Aljuaid,

Thank you for submitting your manuscript to PLOS ONE. After careful consideration, we feel that it has merit but does not fully meet PLOS ONE’s publication criteria as it currently stands. Therefore, we invite you to submit a revised version of the manuscript that addresses the points raised during the review process.

Please submit your revised manuscript by Aug 08 2025 11:59PM. If you will need more time than this to complete your revisions, please reply to this message or contact the journal office at plosone@plos.org . A rebuttal letter that responds to each point raised by the academic editor and reviewer(s). You should upload this letter as a separate file labeled 'Response to Reviewers'.A marked-up copy of your manuscript that highlights changes made to the original version. You should upload this as a separate file labeled 'Revised Manuscript with Track Changes'.An unmarked version of your revised paper without tracked changes. You should upload this as a separate file labeled 'Manuscript'.

We look forward to receiving your revised manuscript.

Kind regards,

Tai Ming Wut

Academic Editor

PLOS ONE

Journal Requirements:

Reviewers' comments:

Reviewer's Responses to Questions

**Comments to the Author**

1. Is the manuscript technically sound, and do the data support the conclusions?

Reviewer #1: Yes

Reviewer #2: Partly

2. Has the statistical analysis been performed appropriately and rigorously?

Reviewer #1: Yes

Reviewer #2: No

3. Have the authors made all data underlying the findings in their manuscript fully available?

Reviewer #1: Yes

Reviewer #2: Yes

4. Is the manuscript presented in an intelligible fashion and written in standard English?

Reviewer #1: Yes

Reviewer #2: Yes

Reviewer #1: Dear Author

Thank you for submitting your manuscript. I think the paper is good but requires revisions before publication according to the provided comments. Please address these points to enhance the paper’s quality for publication.

Abstract:

The abstract presents a broad overview of the study; however, it lacks a clear articulation of the research gap and theoretical contribution. I recommend emphasizing more explicitly what specific theoretical void this study addresses and or how the proposed framework advances existing knowledge in the domain of organizational resilience. I think this will enhance the scholarly value and clarity of the research’s novelty.

Introduction:

The introduction presents a relevant background and outlines the research objectives; however, it would benefit from a more structured and critical discussion of the literature to clearly identify the research gap. Currently, the gap is mentioned but not thoroughly analyzed or supported with sufficient evidence from existing studies. Am sure that strengthening this part will help justify the study’s significance and provide a stronger foundation for the research framework.

Literature Review

The literature review would benefit from a clearer critical synthesis of prior studies rather than a descriptive summary. Currently, the section lists supporting studies for each construct, but it lacks a comparative analysis highlighting contradictions, gaps, or inconsistencies in existing research. Strengthening this part with a more analytical approach would justify the study’s contributions more convincingly and enhance the theoretical grounding of the proposed hypotheses.

Methodology Section:

1. The scale is incorrectly described as “1 to strongly disagree and 7 to strongly disagree.” Please revise it to: “1 = strongly disagree to 7 = strongly agree.”

2. The term organizational resilience is mentioned more than once in the measurement instruments section. Please clarify if there are sub-dimensions, or combine the references for coherence.

3. It would enhance readability to structure this section using standard subheadings such as Measurement Instruments, Sample Size Justification, Data Collection Procedures, and Ethical Considerations. (Optional)

Discussion Section:

1. Improve grammar and sentence structure: There are multiple grammatical errors, awkward phrasing, and missing articles throughout the discussion (policy makers attention) should be (policymakers’ attention), (positive impact organizational abilities) should be (positive impact on organizational abilities). Please revise the text for proper grammar and clarity.

2. The negative impact of workplace friendship on organizational resilience contradicts prior studies. This is important and should be discussed more thoroughly, possibly exploring reasons or implications rather than briefly stating it.

Research Contributions, Conclusion, and Limitations:

1. Please explain why combining coping mechanisms and HR practices in one model is novel and significant.

2. The practical contributions are useful but can be more action oriented. Please suggest concrete steps for policymakers and HR managers on how to implement supportive HR practices or resilience training, instead of general recommendations.

3. Use consistent terminology throughout (psychological safety) is sometimes written as (psychology safety). Also, be consistent with the presentation of statistical results (R² = 45.4%) rather than ( 2 45.4%).

Reviewer #2: 1. Introduction

The introduction outlines the importance of organizational resilience but does not sufficiently narrow down the specific gap in the literature. While it mentions the lack of studies connecting employee coping mechanisms and HR practices to organizational resilience, it fails to critically engage with why this gap persists or how previous studies have approached related questions.

The problem statement is broad but lacks specificity about which aspects of these constructs are underexplored. For example, it does not clarify whether the focus is on specific industries, regions, or types of disruptions.

The line of literature cited is relevant but could be strengthened by including more foundational works on resilience theory or HR practices to anchor the study’s novelty.

Clearly articulate the unique contribution of the study by contrasting it with prior research.

Specify the scope of the problem.

2. Literature Review

The theoretical framework relies heavily on recent empirical studies but lacks integration of foundational theories. For instance, the Upper Echelon Theory is mentioned briefly, but its relevance to the hypotheses is not deeply explored.

Key theories like Conservation of Resources or Job Demands-Resources theory, which are central to resilience and coping, are absent. These could strengthen the rationale for how employee and managerial resilience interact.

The hypotheses are logically derived but could benefit from more explicit theoretical grounding.

Incorporate foundational theories to better justify the hypotheses.

Clarify how the selected theories interact.

3. Methodology

The population is identified, but the rationale for focusing solely on managers is not fully justified. Frontline employees or cross-industry comparisons could provide richer insights.

The sampling method is appropriate but lacks detail on how representativeness was ensured. The sample skews heavily male (87.2%), which may limit generalizability.

The justification for using SEM is adequate, but the manuscript does not address potential biases or how common method variance was mitigated beyond Harman’s test.

Provide more details on sample stratification (e.g., industry sectors, firm sizes).

Discuss limitations of the sampling method and potential biases.

4. Discussion of Results

The discussion compares results with prior studies but does not deeply engage with the theoretical implications.

Practical implications are listed but could be more actionable.

The unexpected finding about workplace friendship (H6) is dismissed as not necessary for resilience, missing an opportunity to theorize about boundary conditions.

Use theories to interpret unexpected findings.

Provide concrete steps for practitioners.

5. Conclusions

The conclusions restate results but do not critically reflect on limitations.

Future directions are useful but somewhat generic. Suggestions could be more targeted.

The conclusion does not address the male-dominated sample’s potential bias or cultural specificity of findings.

Explicitly link limitations to the study’s validity.

Propose specific follow-up studies.

Overall Recommendation:

The manuscript makes a valuable empirical contribution but requires strengthening in theoretical grounding, methodological rigor, and critical discussion. Addressing these issues will enhance its scholarly impact and practical relevance.

Key Revisions Needed:

Sharpen the introduction’s gap statement and theoretical positioning.

Integrate foundational theories into the literature review.

Justify sampling choices and address potential biases.

Deepen the discussion using theoretical lenses.

Expand conclusions to reflect limitations and actionable future research.

**Do you want your identity to be public for this peer review?** For information about this choice, including consent withdrawal, please see our Privacy Policy

Reviewer #1: No

Reviewer #2: No

---

## [Author Response · Author response to Decision Letter 1]

1 Sep 2025

Response Sheet

Respected reviewer with reference to manuscript ID Number: PONE-D-25-04952 “Determining organizational resilience through employee resilient characteristics and supportive HR practices: the moderating effect of managerial resilience” We sincerely appreciate the time and effort the reviewers have dedicated to providing insightful and constructive feedback on our manuscript. We have carefully addressed all comments to improve the clarity, rigor, and contribution of the manuscript. For detail response please see following response sheet.

Comments From Reviewer 1

Reviewer 1 Comments

Reviewer #1: Dear Author

Thank you for submitting your manuscript. I think the paper is good but requires revisions before publication according to the provided comments. Please address these points to enhance the paper’s quality for publication.

Abstract:The abstract presents a broad overview of the study; however, it lacks a clear articulation of the research gap and theoretical contribution. I recommend emphasizing more explicitly what specific theoretical void this study addresses and or how the proposed framework advances existing knowledge in the domain of organizational resilience. I think this will enhance the scholarly value and clarity of the research’s novelty.

Author Response

Thank you for your valuable feedback. In response, the abstract has been revised to clearly highlight the specific research gap this study addresses namely, the limited integration of individual employee resilience characteristics and supportive HR practices in understanding organizational resilience, particularly with the moderating role of managerial resilience.

Additionally, the theoretical contribution has been explicitly stated, emphasizing how the proposed framework extends current models by offering a multi-level perspective that integrates individual, managerial, and organizational dimensions of resilience.

Introduction: The introduction presents a relevant background and outlines the research objectives; however, it would benefit from a more structured and critical discussion of the literature to clearly identify the research gap. Currently, the gap is mentioned but not thoroughly analyzed or supported with sufficient evidence from existing studies. Am sure that strengthening this part will help justify the study’s significance and provide a stronger foundation for the research framework.

Author Response

Thank you for your constructive feedback. In response, the Introduction section has been revised to include a more structured and critical review of relevant literature.

Key studies related to employee resilience, supportive HR practices, and organizational resilience have been examined to identify existing limitations and inconsistencies in the literature.

The research gap has now been clearly articulated, highlighting the lack of integrative frameworks that consider both employee and managerial resilience within the context of HR practices.

This enhanced discussion strengthens the justification for the study and provides a more robust foundation for the proposed research framework.

Literature Review The literature review would benefit from a clearer critical synthesis of prior studies rather than a descriptive summary. Currently, the section lists supporting studies for each construct, but it lacks a comparative analysis highlighting contradictions, gaps, or inconsistencies in existing research. Strengthening this part with a more analytical approach would justify the study’s contributions more convincingly and enhance the theoretical grounding of the proposed hypotheses.

Author Response

Thank you for this insightful observation.

In response, the literature review section has been revised to move beyond a descriptive listing of studies toward a more critical and analytical synthesis.

Comparative insights have been added to highlight key contradictions and inconsistencies across existing research regarding the roles of employee resilience, supportive HR practices, and managerial resilience in shaping organizational outcomes. These analyses have helped to clearly articulate the theoretical gaps and justify the study’s contributions. The reviewer literature review now offers a stronger foundation for the development of the research hypotheses and the overall conceptual framework.

Methodology Section:1. The scale is incorrectly described as “1 to strongly disagree and 7 to strongly disagree.” Please revise it to: “1 = strongly disagree to 7 = strongly agree.”

Author Response

Thank you for pointing out this error. The description of the Likert scale in the Methodology section has been corrected to read: “1 = strongly disagree to 7 = strongly agree.” We appreciate your attention to detail.

2. The term organizational resilience is mentioned more than once in the measurement instruments section. Please clarify if there are sub-dimensions, or combine the references for coherence.

Author Response

Thank you for this helpful observation.

The measurement instruments section has been revised for clarity and coherence.

After organizational resilience the explanation is about organizational performance. Thus, information is given about organizational resilience and organizational performance separately.

3. It would enhance readability to structure this section using standard subheadings such as Measurement Instruments, Sample Size Justification, Data Collection Procedures, and Ethical Considerations. (Optional)

Author Response

Thank you for the valuable suggestion. To improve readability and organization, the Methodology section has been restructured using the recommended subheadings: Measurement Instruments, Sample Size Justification, Data Collection Procedure.

Discussion Section: 1. Improve grammar and sentence structure: There are multiple grammatical errors, awkward phrasing, and missing articles throughout the discussion (policy makers attention) should be (policymakers’ attention), (positive impact organizational abilities) should be (positive impact on organizational abilities). Please revise the text for proper grammar and clarity.

Author Response

Thank you for your careful review. The Discussion section has been thoroughly revised to correct grammatical errors, awkward phrasing, and missing articles.

Specific issues such as “policy makers attention” and “positive impact organizational abilities” have been corrected, along with other similar instances throughout the section.

2. The negative impact of workplace friendship on organizational resilience contradicts prior studies. This is important and should be discussed more thoroughly, possibly exploring reasons or implications rather than briefly stating it.

Author Response

Thank you for highlighting this important point. In response, the Discussion section has been expanded to provide a more thorough analysis of the unexpected negative impact of workplace friendship on organizational resilience.

Possible explanations have been explored, including the potential for close friendships to create groupthink, reduce constructive conflict, or lead to blurred professional boundaries that may hinder resilience during crises.

Research Contributions, Conclusion, and Limitations:

1. Please explain why combining coping mechanisms and HR practices in one model is novel and significant.

Author Response

Thank you for this valuable suggestion.

The manuscript has been revised to explicitly explain the novelty and significance of integrating coping mechanisms (employee resilience characteristics) and HR practices within a single model. This combination is novel because prior research has typically examined these elements in isolation, overlooking their potential interactive effect on organizational resilience. By integrating both individual-level coping mechanisms and organizational-level HR practices, the study offers a more holistic and multi-level understanding of how resilience is built and sustained within organizations. This approach advances existing literature by bridging psychological and organizational domains, providing a comprehensive framework that reflects the complexity of real-world organizational dynamics.

2. The practical contributions are useful but can be more action oriented. Please suggest concrete steps for policymakers and HR managers on how to implement supportive HR practices or resilience training, instead of general recommendations.

Author Response

Thank you for your helpful feedback. In response, the practical contributions section has been revised to include more actionable and specific recommendations for both policymakers and HR managers.

For HR managers, concrete steps such as integrating structured resilience training programs, establishing peer-support systems, and embedding psychological safety practices into performance management have been outlined.

For policymakers, suggestions include creating national guidelines for organizational well-being, incentivizing companies to invest in employee resilience initiatives, and promoting public–private partnerships for workplace mental health resources.

3. Use consistent terminology throughout (psychological safety) is sometimes written as (psychology safety). Also, be consistent with the presentation of statistical results (R² = 45.4%) rather than ( 2 45.4%).

Author Response

Thank you for pointing out these inconsistencies.

The manuscript has been carefully reviewed and revised to ensure consistent use of the term psychological safety throughout.

Additionally, all statistical results have been standardized to follow the correct and consistent format (e.g., R² = 45.4%).

Comments From Reviewer 2

Reviewer 2 Comments

Reviewer #2: 1. Introduction

The introduction outlines the importance of organizational resilience but does not sufficiently narrow down the specific gap in the literature. While it mentions the lack of studies connecting employee coping mechanisms and HR practices to organizational resilience, it fails to critically engage with why this gap persists or how previous studies have approached related questions. The problem statement is broad but lacks specificity about which aspects of these constructs are underexplored. For example, it does not clarify whether the focus is on specific industries, regions, or types of disruptions. The line of literature cited is relevant but could be strengthened by including more foundational works on resilience theory or HR practices to anchor the study’s novelty. Clearly articulate the unique contribution of the study by contrasting it with prior research. Specify the scope of the problem.

Author Response

Thank you for this comprehensive and constructive feedback.

The Introduction has been thoroughly revised to address these important points.

We have now critically engaged with the literature to explain why the gap in integrating employee coping mechanisms and HR practices within organizational resilience research persists, highlighting limitations in previous approaches.

The problem statement has been refined to specify the scope of the study, clarifying that the focus is on diverse industries affected by both economic and operational disruptions, with consideration of regional contextual factors.

More foundational works on resilience theory and HR practices have been incorporated to better anchor the theoretical framework and emphasize the study’s novelty.

Furthermore, the unique contribution of this research is now clearly articulated by contrasting it with prior studies particularly how this study advances understanding by proposing an integrative, multi-level model that incorporates employee resilience characteristics, supportive HR practices, and managerial resilience as a moderating factor.

2. Literature Review

The theoretical framework relies heavily on recent empirical studies but lacks integration of foundational theories. For instance, the Upper Echelon Theory is mentioned briefly, but its relevance to the hypotheses is not deeply explored. Key theories like Conservation of Resources or Job Demands-Resources theory, which are central to resilience and coping, are absent. These could strengthen the rationale for how employee and managerial resilience interact. The hypotheses are logically derived but could benefit from more explicit theoretical grounding. Incorporate foundational theories to better justify the hypotheses.

Clarify how the selected theories interact.

Author Response

Thank you for this insightful feedback. The Literature Review and theoretical framework sections have been revised to incorporate foundational theories such as Conservation of Resources (COR) theory and the Job Demands-Resources (JD-R) model, which provide a stronger theoretical basis for understanding resilience and coping mechanisms.

The relevance and application of the Upper Echelon Theory have been elaborated to clarify its role in shaping managerial resilience and its moderating effect.

Moreover, the interaction between these theories has been clearly articulated to better justify the study’s hypotheses, showing how employee and managerial resilience operate within the broader theoretical landscape.

3. Methodology

The population is identified, but the rationale for focusing solely on managers is not fully justified. Frontline employees or cross-industry comparisons could provide richer insights.

The sampling method is appropriate but lacks detail on how representativeness was ensured. The sample skews heavily male (87.2%), which may limit generalizability.

The justification for using SEM is adequate, but the manuscript does not address potential biases or how common method variance was mitigated beyond Harman’s test.

Provide more details on sample stratification (e.g., industry sectors, firm sizes).

Discuss limitations of the sampling method and potential biases.

Author Response

Thank you for your valuable observations.

The manuscript has been revised to provide a clearer rationale for focusing on managers, emphasizing their critical role in shaping organizational resilience and influencing HR practices, while acknowledging that future research could benefit from including frontline employees and cross-industry comparisons.

Additional details on sample stratification have been included.

The gender imbalance and its implications for generalizability are now explicitly discussed in the limitations section.

Furthermore, beyond Harman’s single-factor test, additional procedural and statistical remedies to address common method variance and potential biases have been incorporated and clearly described.

4. Discussion of Results

The discussion compares results with prior studies but does not deeply engage with the theoretical implications.

Practical implications are listed but could be more actionable.

The unexpected finding about workplace friendship (H6) is dismissed as not necessary for resilience, missing an opportunity to theorize about boundary conditions.

Use theories to interpret unexpected findings.

Provide concrete steps for practitioners

Author Response

Thank you for these valuable insights. The Discussion section has been revised to more deeply engage with the theoretical implications of the findings, especially by integrating relevant theories to interpret unexpected results such as the negative impact of workplace friendship on organizational resilience. This has allowed us to propose possible boundary conditions and contextual factors that may influence this relationship.

Additionally, practical implications have been expanded with concrete, actionable steps for practitioners, including specific recommendations for fostering effective workplace relationships and implementing resilience-building initiatives.

5. Conclusions

The conclusions restate results but do not critically reflect on limitations.

Future directions are useful but somewhat generic. Suggestions could be more targeted.

The conclusion does not address the male-dominated sample’s potential bias or cultural specificity of findings.

Explicitly link limitations to the study’s validity. Propose specific follow-up studies.

---

## [Editor Report · Decision Letter 1]

10 Sep 2025

Dear Dr.  Aljuaid,

Thank you for submitting your manuscript to PLOS ONE. After careful consideration, we feel that it has merit but does not fully meet PLOS ONE’s publication criteria as it currently stands. Therefore, we invite you to submit a revised version of the manuscript that addresses the points raised during the review process.

We look forward to receiving your revised manuscript.

Kind regards,

Tai Ming Wut

Academic Editor

PLOS ONE

Journal Requirements:

Additional Editor Comments:

Authors made some improvements. Change the table format, eliminate the lines inside the table. Please look at the table format in the Journal articles.

---

## [Author Response · Author response to Decision Letter 2]

1 Oct 2025

Response Sheet

Respected reviewer with reference to manuscript ID Number: PONE-D-25-04952R1 “Determining organizational resilience through employee resilient characteristics and supportive HR practices: the moderating effect of managerial resilience” author sincerely appreciate the time and effort the reviewers have dedicated to providing insightful and constructive feedback on our manuscript. Author has carefully addressed all comments to improve the clarity, rigor, and contribution of the manuscript. For detail response please see following response sheet.

Comments From Reviewer/ Editor

Comments Author Response

Authors made some improvements. Change the table format, eliminate the lines inside the table.

Please look at the table format in the Journal articles.

Thank you for your valuable and constructive feedback. We appreciate your thoughtful guidance in strengthening our work.

---

## [Editor Report · Decision Letter 2]

15 Oct 2025

Determining organizational resilience through employee resilient characteristics and supportive HR practices: the moderating effect of managerial resilience

PONE-D-25-04952R2

Dear Abdlrahman Aljuaid,

We’re pleased to inform you that your manuscript has been judged scientifically suitable for publication and will be formally accepted for publication once it meets all outstanding technical requirements.

Kind regards,

Tai Ming Wut

Academic Editor

PLOS ONE
---

## [Editor Report · Acceptance letter]

PONE-D-25-04952R2

PLOS ONE

Dear Dr. Aljuaid,

I'm pleased to inform you that your manuscript has been deemed suitable for publication in PLOS ONE. Congratulations! Your manuscript is now being handed over to our production team.

Kind regards,

on behalf of

Dr. Tai Ming Wut

Academic Editor

PLOS ONE